# High-resolution awake mouse fMRI at 14 tesla

**David Hike[1†], Xiaochen Liu[1†], Zeping Xie[1], Bei Zhang[1], Sangcheon Choi[1], Xiaoqing Alice Zhou[1], Andy Liu[1,2], Alyssa Murstein[1,2], Yuanyuan Jiang[1], Anna Devor[1,3], Xin Yu[1]\***

[1]Athinoula A. Martinos Center for Biomedical Imaging, Department of Radiology, Harvard Medical School, Massachusetts General Hospital, Charlestown, United States; [2]Graduate Program in Neuroscience, Boston University, Boston, United States; [3]Department of Biomedical Engineering, Boston University, Boston, United States

## eLife Assessment

This is a **valuable** study describing an implementation of awake mouse fMRI with implanted head coils at high fields. The evidence presented is **convincing**, combining technical advances with interesting neuroscience applications showing that mice anticipate stimuli given at regular (but at irregular) intervals.

**\*For correspondence:**
xyu9@mgh.harvard.edu

[†]These authors contributed equally to this work

**Abstract** High-resolution awake mouse functional magnetic resonance imaging (fMRI) remains challenging despite extensive efforts to address motion-induced artifacts and stress. This study introduces an implantable radio frequency (RF) surface coil design that minimizes image distortion caused by the air/tissue interface of mouse brains while simultaneously serving as a headpost for fixation during scanning. Furthermore, this study provides a thorough acclimation method used to accustom animals to the MRI environment minimizing motion-induced artifacts. Using a 14 T scanner, high-resolution fMRI enabled brain-wide functional mapping of visual and vibrissa stimulation at 100 μm×100 μm×200 μm resolution with a 2 s per frame sampling rate. Besides activated ascending visual and vibrissa pathways, robust blood oxygen level-dependent (BOLD) responses were detected in the anterior cingulate cortex upon visual stimulation and spread through the ventral retrosplenial area (VRA) with vibrissa air-puff stimulation, demonstrating higher-order sensory processing in association cortices of awake mice. In particular, the rapid hemodynamic responses in VRA upon vibrissa stimulation showed a strong correlation with the hippocampus, thalamus, and prefrontal cortical areas. Cross-correlation analysis with designated VRA responses revealed early positive BOLD signals at the contralateral barrel cortex (BC) occurring 2 s prior to the air-puff in awake mice with repetitive stimulation, which was not detected using a randomized stimulation paradigm. This early BC activation indicated a learned anticipation through the vibrissa system and association cortices in awake mice under continuous exposure of repetitive air-puff stimulation. This work establishes a high-resolution awake mouse fMRI platform, enabling brain-wide functional mapping of sensory signal processing in higher association cortical areas.

## Introduction

Functional magnetic resonance imaging (fMRI) indirectly measures brain activity via MRI contrast associated with endogenous blood oxygen level-dependent (BOLD) signals (*Ogawa et al., 1990a*; *Ogawa et al., 1990b*; *Logothetis et al., 2001*). The BOLD contrast was first described by *Pauling*

*and Coryell, 1936*, but it had not been utilized in anesthetized rodent MRI until 1990 (*Ogawa et al., 1990a*; *Ogawa et al., 1990b*). The power of BOLD-fMRI was later revealed in human brain functional mapping (*Ogawa et al., 1992*; *Kwong et al., 1992*; *Bandettini et al., 1992*) and has revolutionized cognitive neuroscience. In contrast to human studies, preclinical fMRI has played a crucial role in method development and validation (*Logothetis et al., 2001*; *Zhou et al., 2023*; *Yu et al., 2016*; *Yoshida et al., 2016*; *Pérez-Cervera et al., 2018*; *Pirttimäki et al., 2016*; *Arbabi et al., 2022*; *Labbé et al., 2021*). fMRI of anesthetized rodents reduces confounding artifacts due to motion and detects robust BOLD or cerebral blood volume signals under various anesthetics (*Masamoto et al., 2007*; *Kawazoe et al., 2022*; *Bukhari et al., 2018*; *Steiner et al., 2021*; *Shim et al., 2018*; *Jonckers et al., 2014*; *Tsurugizawa and Yoshimaru, 2021*; *Bukhari et al., 2017*; *Grandjean et al., 2014*; *Becq et al., 2020*; *Conzen et al., 1992*; *Magnuson et al., 2014*; *Zhao et al., 2008*; *Chen et al., 2019b*; *Wang et al., 2018b*). Recently, the bridging power of preclinical fMRI for basic mechanistic and translational studies has been further exploited given the combination of rodent fMRI with genetic modification tools (e.g. optogenetics, chemogenetics, and genetically encoded biosensors) (*Zhou et al., 2023*; *Jung et al., 2021*; *Cover et al., 2021*; *Rocchi et al., 2022*; *Lee et al., 2022b*; *Zerbi et al., 2019*; *Giorgi et al., 2017*; *Schulz et al., 2012*; *Ioanas et al., 2022*; *Jung, 2022*; *Chen et al., 2019a*; *Lee et al., 2022a*; *Nakamura et al., 2020*; *Peeters et al., 2020*; *Oyarzabal et al., 2022*). Among the many efforts in anesthetized rodent fMRI, mouse fMRI set a foundation for mechanistic multi-modal imaging given its global mapping scheme in genetic modification models (*Kawazoe et al., 2022*; *Shim et al., 2018*; *You et al., 2021*; *Lake et al., 2020*), as well as the ability to perform viral transfections to circuit- or cellular-specific targets in transgenic models. However, anesthetics alter brain function during fMRI, preventing accurate interpretation of brain functional changes in awake states (*Masamoto et al., 2007*; *Jonckers et al., 2014*; *Grandjean et al., 2014*; *Conzen et al., 1992*; *Magnuson et al., 2014*; *Desai et al., 2011*; *Chen et al., 2020*; *Sharp et al., 2015*; *Low et al., 2016a*; *Gargiulo et al., 2012*; *Scheller et al., 1988*; *Crawford et al., 1992*).

Awake mouse fMRI presents itself to provide the most relevant brain functional mapping information for translational cross-scale brain dynamic studies. To immobilize the mouse head during scanning, surgical implantation of headposts has been developed for head-fixation similar to optical imaging schemes (*Yoshida et al., 2016*; *Desjardins et al., 2019*; *Shih et al., 2014*). In contrast to the fMRI mapping of anesthetized animals, motion-induced artifacts and potential stress-related issues caused by loud gradient noises and micro-vibrations during scanning are major difficulties faced by existing awake mouse fMRI studies (*Desai et al., 2011*; *Chen et al., 2020*; *Gutierrez-Barragan et al., 2022*; *Han et al., 2019*; *Xu et al., 2022*; *Liu et al., 2020*; *Almeida et al., 2022*; *Zeng et al., 2022*). Previous work has demonstrated that well-planned training procedures could acclimate awake mice during scanning (*Han et al., 2019*; *Madularu et al., 2017b*; *Xu et al., 2022*; *Liu et al., 2020*; *Almeida et al., 2022*; *Low et al., 2016b*; *Chiba et al., 2012*; *Ferenczi et al., 2016*); however, different training paradigms are expected to produce large variability in the functional mapping results (*Mandino et al., 2024*). One ongoing challenge of awake mouse fMRI is to provide reproducible and high-quality brain functional images with sufficient spatiotemporal resolution and signal-to-noise ratio (SNR) to distinguish functional nuclei of only a few hundred microns in mouse brains. Since increasing spatiotemporal resolution leads to a reduction in SNR of the images, accessing the highest field MRI available, as well as maximizing the efficiency of the radio frequency (RF) transceiver signal is critical. Although cryoprobes have been well implemented to boost SNR (*Yoshida et al., 2016*; *Chen et al., 2020*; *Niendorf et al., 2015*), construction limitations of the superconducting environment constrain the usable space and flexibility to accommodate other imaging/recording modalities (*Yoshida et al., 2016*; *Arbabi et al., 2022*; *Baltes et al., 2009*; *Wright et al., 2000*; *Kwok and You, 2006*; *Takata et al., 2015*; *Abe et al., 2021*; *Hamada, 2024*). Implantable coils have been used in animal imaging for over three decades (*Pirttimäki et al., 2016*; *Farmer et al., 1990*; *Summers et al., 1995*; *Logothetis et al., 2002*; *Wang et al., 2018a*; *Lee et al., 2024*; *Madularu et al., 2017a*; *Chen et al., 2022*). Their use gained popularity due to higher SNR and reduction of susceptibility artifacts. The main limitation of implantable coils is the need to surgically implant these coils, adding a degree of invasiveness that MRI usually avoids. However, for typical awake mouse neuroimaging studies, surgical procedures to provide a head-fixation apparatus are routinely practiced. Replacing the conventional headpost for immobilization of the head with an implantable RF coil is critical for achieving high-resolution awake mouse fMRI using ultra-high field MRI, e.g., 14 T shown here.

In this present study, we established an awake mouse fMRI platform by applying an implantable RF surface coil, permanently affixed to the head, which simultaneously functioned as a headpost for fixation during scanning, minimizing animal motion. This setup allowed us to acquire images with an in-plane spatial resolution of 100 μm and 200 μm slice thickness. This unique implantable RF coil/headpost scheme simplified the awake mouse training and conditioning for imaging. While there is currently insufficient evidence to ascertain whether head-fixed training leads to stress-free animals, we observed that a 5-week training scheme resulted in increased eye movements, presenting decreased struggling and freezing behavior indicative of calmer awake mice during scanning. This implanted RF coil scheme also improved $B_0$ homogeneity, as well as effectively eliminated any motion-related loading changes causing $B_1$ variability. Here, we successfully mapped activated visual and vibrissa pathways and detected robust BOLD responses in higher-order association cortices, e.g., anterior cingulate area (ACA) with visual stimulation and ventral retrosplenial area (VRA) with vibrissa stimulation in awake mice based on connectivity map projections from the Allen Brain Atlas derived from a Cre-dependent AAV tracing of axonal projections (*Oh et al., 2014*). Interestingly, the repetitive vibrissa stimulation paradigm in awake mice has enabled us to detect potential anticipatory learning with mice predicting the onset of stimulation. Our work is a fundamental step toward combining high-resolution fMRI with other modalities to simultaneously record neuronal and microvascular signals throughout brain-wide circuity in awake mice.

## Methods

### Animals

Thirty-eight C57BL/6 mice were used in the current study (weighing between 20 g and 30 g) and allocated as follows for each experiment: SNR measurements at 9.4 T – 16 male mice; SNR, visual, and whisker stimulation measurements at 14 T – 13 mice (6F/7M); random stimulation measurements at 14 T – 9 mice (4F/5M). Mice were group-housed (3–4/cage) under a 12 hr light/dark cycle with food and water ad libitum. All animal procedures were conducted in accordance with protocols approved by the Massachusetts General Hospital (MGH) Institutional Animal Care and Use Committee (IACUC) under protocol number 2020N000073, and animals were cared for according to the requirements of the National Research Council's Guide for the Care and Use of Laboratory Animals.

### Awake mouse fMRI setup

The awake mouse cradle was designed in Blender (Blender Foundation, Amsterdam, NL) and 3D printed using a Formlabs 3L 3D printer (Formlabs Inc, Somerville, MA, USA). The design incorporated a sliding track which accepted the printed circuit board (PCB) chip transceiver circuit to slide in while the mouse was inserted into the cradle. Two transceiver circuit designs were built, a single loop and a 'figure 8' design. Each one keeps the $B_1$ direction orthogonal to the $B_0$. The single loop allows for full brain coverage at sufficient depths for subcortical investigation. The 'figure 8' design, due to its smaller coil loops and $B_1$ direction, is limited to precise measurements of shallow brain regions but provides a significant increase in SNR which is beneficial to cortical-specific studies which do not have a need to look deeper into subcortical regions but would benefit from a much higher SNR. The single loop or 'figure 8' shape RF coils were built to optimize tuning/matching performance when affixed onto the mouse skull. The coils serve to optimize the $B_0$ homogeneity by minimizing the air-tissue interface. Each coil was built to weigh ~2.5 g minimizing the recovery/neck strengthening time of each mouse. The standardized RF coil was acquired from MRIBOT LLC (Malden, MA, USA) using the circuit diagram shown in *Figure 1—figure supplement 1*.

### Animal surgery

Mice underwent surgery to affix the RF coil to the head. Animals were anesthetized for surgery using isoflurane. Induction was accomplished using 5% isoflurane and 1 L/min of medical air and 0.2 L/min additional $O_2$ flow. Animals were maintained at 1.5–2% isoflurane using respiration rate as a monitor for anesthesia depth. To attach the head coil, mice were affixed in a stereotaxic stage to stabilize the head with ear and bite bars. The scalp was shaved sterilized with ethanol and iodine and an incision was made to expose an area of the skull the size of the RF coil ring. The skull was cleared of residual tissue and cleaned with 0.3% $H_2O_2$ and PBS before being dried. The coil was then positioned over the

skull covering the underlying brain. The coil ring was lifted ~0.3–0.5 mm above the surface of the skull to avoid over-loading effects and held in place while a thin layer of cyanoacrylate glue was applied to connect the coil to the skull. Once dried (~5–8 min), two-part dental cement (Stoelting Co., Wood Dale, IL, USA) was mixed and applied to cover the coil and exposed bone paying special note to the base of the coil to firmly secure it and avoid air bubbles and drips toward the eyes. The edges of the skin were then glued to close the surgical site. After the dental cement had fully hardened (~10 min), the mouse was released from the stereotaxic stage and received subcutaneous injections of dexamethasone and cefazolin. Mice were then allowed to recover in their home cage for at least 1 week to ensure ample neck strengthening had occurred and the mice could walk with normal head posture.

## Animal training

To acclimate the animals to the MRI environment, each mouse underwent 5 weeks of intermittent habituation procedures to train animals before fMRI experiments by using the following method: Training days pre-surgery (Phase 1)

1. in hand mouse handling → 5 min
2. in hand mouse handling → 10 min

Holder training days post-surgery (after recovery) (Phase 2)

1. secured in holder → 15 min
   a. 10 min pupil recording
2. secured in holder → 30 min
   a. 10 min pupil recording

Mock-MRI training days (Phase 3)

1. secured in holder with MRI audio → 30 min
   a. 10 min pupil recording
2. secured in holder with MRI audio → 30 min
   a. 10 min pupil recording
3. secured in holder with MRI audio → 30 min
   a. 10 min pupil recording
4. secured in holder with MRI audio → 30 min
   a. 10 min pupil recording
5. secured in holder with MRI audio → 60 min
   a. 10 min pupil recording
6. secured in holder with MRI audio → 60 min
   a. 10 min pupil recording
7. secured in holder with MRI audio → 60 min
   a. 10 min pupil recording after
8. secured in holder with MRI audio → 60 min
   a. 10 min pupil recording

Training days inside MRI (resting-state and stimulation) (Phase 4)

**Table 1.** Table showing the training paradigm for each of the four phases of training.
rs → resting-state fMRI, stim → whisker stimulation fMRI.

| Training day of phase | Phase 1 (hold in hand) | Phase 2 (holder+pupil) | Phase 3 (mock-MRI+pupil) | Phase 4 (EPI+pupil) |
|---|---|---|---|---|
| 1 | 5 min | 15 min | 30 min | 60 min (rs) |
| 2 | 10 min | 30 min | 30 min | 60 min (rs) |
| 3 | | | 30 min | 60 min (stim) |
| 4 | | | 30 min | 60 min (stim) |
| 5 | | | 60 min | |
| 6 | | | 60 min | |
| 7 | | | 60 min | |
| 8 | | | 60 min | |

1. Real scans (EPI with pupil recording)
2. Real scans (EPI with pupil recording)
3. Real scans (EPI with air-puff and pupil recording)
4. Real scans (EPI with air-puff and pupil recording)

Following this, data acquisition began, and pupil changes were monitored during scans. During all pupil recordings, animals were secured in the holder and ensured the environment for recording did not allow external light to reach the pupil. Illumination was achieved from a 660 nm LED light source (Thorlabs, Inc, Newton, NJ, USA) delivered via fiber-optic cable. Videos were captured at 30 fps using a 1/3" CMOS camera and 12 mm focal length lens (Tru Components, Chicago, IL, USA).

## Pupil/eye fluctuations during training regime

The pupils of awake mice were recorded during training sessions, allowing investigation into eye movements and pupil diameter changes as potential surrogate of stress-related readouts of the animals. The pupil recordings were measured over 12 training days across 3–4 weeks in Phases 2–4 (*Table 1*). In Phase 1, animals were gently held in hands, not head-fixed in the cradle. In the second half of Phase 4, head-fixed animals were exposed to air-puff stimulation inside the MR scanner during echo planar imaging (EPI) sequences acquisition. At this stage, animals tended to close their eyes in response to air-puff, so pupil measurements were not feasible to be included for data analysis. The increased eye movements were well detected in Phase 3 when animals were exposed to the real MRI acoustic noise in the head-fixed position (located in the RF shielded box attached to the 14 T magnet, i.e. the CCM box) (*Figure 2—figure supplement 1A*). Interestingly, in Phase 4, animals showed significantly reduced eye movements during the first day when animals were positioned inside the MR scanner with EPI scanning, while an increase in eye movement went back to the level of Phase 3 in the following training days. The pupil diameter changes were also measured as the function of training days. The power spectral analysis showed ultra-slow pupil dynamic changes with peaked bandwidths less than 0.02 Hz. Interestingly, the power of the ultra-slow pupil dynamics also increased as the function of time similar to eye movements, in particular, during Phase 3. Meanwhile, power reduction in the first day of in-bore training followed with recovered pupil dynamics in the following days was also observed during Phase 4. Although the actual stress of the animals during scanning remains to be further investigated following the 5-week training procedure, the motion-induced image distortion has been dramatically reduced in well-trained animals compared to the start of in-bore training.

## Anesthesia regiment for MRI measurements of SNR

While acquiring images to measure SNR improvements, all animals were anesthetized for the duration of MR scanning. Mice were induced using 5% isoflurane in medical air and maintained with 1.0–2.0% isoflurane, adjusted to retain stable physiological conditions while in the magnet. The gas mixture was supplied through the hollow bite bar directly to the mouth and nose of the animal at a rate of 1.0 L/min. Animals were anesthetized to minimize artifacts associated with motion. Physiological monitoring of the animal was performed through the integration of a Small Animal Monitoring and Gating System (Model 1030, SA Instruments, Inc, Stony Brook, NY, USA) capable of recording respiration, body temperature, electrocardiogram, and other parameters. The animal's breathing rate was continuously monitored and recorded during scanning using a pressure-sensitive sensor-pad and maintained between 50 and 80 breath/min. Animals were kept at a constant temperature of 37°C in the MRI scanner by means of blowing warm air through the bore and recorded using a rectal thermometer probe.

## MRI methods

[1]H MRI data was acquired using the 14 T and 9.4 T horizontal MRI scanners (Magnex Sci, UK) located at the Athinoula A. Martinos Center for Biomedical Imaging in Boston, MA. The 14 T magnet is equipped with a Bruker Avance Neo Console (Bruker-Biospin, Billerica, MA, USA) and is operated using ParaVision 360 V.3.3. A microimaging gradient system (Resonance Research, Inc, Billerica, MA, USA) provides a peak gradient strength of 1.2 T/m over a 60 mm diameter. The 9.4 T scanner is equipped with a Bruker Avance III HD Console (Bruker-Biospin, Billerica, MA, USA) and is operated using ParaVision 6. A dual microimaging gradient system comprises a Bruker gradient coil capable of 44 G/cm, and a Resonance Research gradient insert capable of 150 G/cm.

## SNR measurements

[1]H MRI data for SNR measurements were acquired on 9.4 T (400 MHz) and 14 T (600 MHz) scanners using the following parameters for both systems: TE/TR = 3 ms/475 ms, flip angle = 30°, and four averages for an approximate acquisition time of 4.5 min.

9.4 T scanner was only used to show SNR improvements from the implantable coils. The BOLD fMRI data were collected only at 14 T due to the much-improved SNR available and were collected solely in awake mice to investigate signal associated with the awake functional connectivity. Furthermore, the 'figure 8' shape coils were only used to show the SNR improvement due to the coil design for cortical measurements.

## fMRI BOLD imaging

Multi-slice 2D gradient echo EPI was used to acquire fMRI BOLD data from the awake animals with the following parameters: TE/TR = 7 ms/1 s, segments = 2, bandwidth = 277,777 Hz, 100 μm ×100 μm in plane resolution with a 200 μm slice thickness, 36 slices, 205 repetitions for an acquisition time of 6 min 50 s.

## Anatomical imaging

[1]H MRI data for anatomical registration data were acquired using a multi-slice $T_1$-weighted 2D gradient echo fast low angle shot (FLASH) sequence with the same parameters of the SNR measurement scans except the resolution was adjusted to match the BOLD data at 100 μm×100 μm×200 μm resolution with the parameters mentioned in the 'SNR measurements' subsection.

## Stimulation method/paradigm

The visual and vibrissa stimulation block paradigm was designed as follows: 5 baseline scans, 1 stimulation trigger scan, 19 inter-stimulation scans, and 10 epochs. The visual stimulation used two different wavelengths of light: 530 nm and 490 nm, which flashed at 5 Hz and 5.1 Hz, respectively, for 8 s with a 20 ms 'on' time of each illumination. The whisker air-puff stimulation used the same block design as the visual stimulation but was stimulated with a 10 ms puff duration and an 8 Hz firing rate for 8 s. Due to the use of two segments for these experiments, the effective TR was 2 s. Therefore, the stimulation duration for both visual and vibrissa experiments resulted in four consecutive scans being included in the 'on' stimulation period and 16 consecutive scans being included in the 'rest' period (*Figure 3—figure supplement 1*). The random vibrissa stimulation paradigm used the same 10 ms air-puff at 8 Hz for 8 s but randomized the 'rest' duration. 'Rest' durations were 12 s, 22 s, 32 s, and 42 s and randomized in three different sequences maintaining a scan duration of 205 TRs for each experiment.

## Processing/analysis methods (AFNI and MATLAB)

SNR was computed by dividing mean signal over the standard deviation of the noise. SNR line profile signal data was collected using Amira software (Thermo Fisher Scientific Inc, Waltham, MA, USA). fMRI data was processed using Analysis of Functional Neuroimages (AFNI) (*Cox, 1996*; *Cox and Hyde, 1997*). Bruker 2dseq images of the EPI and FLASH scans were converted to AFNI format using 'to3d' before masking and aligning the dataset to a template.

To process the high-resolution stimulated BOLD response from the visual and vibrissa stimulation paradigms, we developed a processing pipeline (*Figure 3—figure supplement 2*). For each experiment, FLASH data were averaged for anatomical localization and the EPI scans were time averaged before registration. The time-averaged EPI was then registered to the FLASH and then to the Australian Mouse Brain Mapping Consortium (AMBMC) atlas (*Janke and Ullmann, 2015*) where a mask was generated. Each time series for each experiment was concatenated so each experiment contains one long time series dataset using the '3dTcat' command. Data were then despiked before each EPI time point was registered, via a 6-degree transformation, to the atlas using the 'volreg' command after which the previously generated mask was applied. The 'blur' command was used to smooth the newly transformed data before it was scaled and underwent a linear regression. All concatenated data was then split and summed, per each experimental study, to undergo motion correction and outlier removal. The corrected data was then summed and averaged with the remaining processed data to generate a single time series across all experiments. A clustering threshold was set at 100 voxels and the Pearson correlation values were limited to p≤0.01 (corrected) with estimated false discover rate at

q=0.00078. For the random stimulation design, the three runs were concatenated into a single time series each ensuring they all followed the same series of random timings.

## Results

### Development and efficiency validation of implantable RF coils to boost SNR

We have developed an implantable RF coil which effectively boosted the SNR in ultra-high field MRI. Here, we compared two prototypes: a simple single loop coil design and a 'figure 8' coil design. These coils were used to check SNR in anatomical data of anesthetized mice at 9.4 T and 14 T. *Figure 1* shows examples of the prototyped RF coils (*Figure 1A*). The acquired SNR values for each prototype design are shown in *Figure 1B*. Here, we used a commercial four phase-array coil (400 MHz for 9.4 T) as a control to compare with the implantable RF coils. The single loop implantable coils improved SNR over 100% compared the commercial option while the 'figure 8' style showed a more than five times increase at 9.4 T in the cortical regions. The SNR along the dorsal-ventral axis was plotted to compare the $B_1$ field sensitivity of the single loop and 'figure 8' RF coils in comparison with the phase-array coil at 9.4 T, showing significantly increased SNR up to 4 mm depth (*Figure 1B*). Moving up to 14 T, the SNR improvements increased proportionally as a factor of field strength (*Pohmann et al., 2016*). This improved SNR allows for high spatial resolution fMRI studies of awake mice. The single loop coil tuned to 14 T (600 MHz) was used for functional data collection in the manuscript. The 'figure 8' coil was only used as part of development to show the improvement of the coil design for cortical MR signal measurements.

### Awake mouse fMRI with visual stimulation

The RF coil was implanted on the mouse skull to serve as an attachment for head-fixation during awake mouse fMRI at 14 T (*Figure 2A*). The awake mouse fMRI setup was designed using a 3D printed cradle incorporating a sliding track which enabled the PCB chip mounted on the mouse head to slide through. The PCB chip was then fixed in place at the end of the cradle using friction screws (*Video 1*). Once the mouse was fixed in the animal cradle, either a mirror or air tube was positioned for pupillometry recording or vibrissa stimulation, respectively. Additionally, an MRI-compatible camera was incorporated to record the pupil dynamic changes and whisking behavior of awake mice during scanning. One key feature of the awake mouse fMRI setup is the plug and play capability for scanning.

This awake mouse fMRI setup enabled high-resolution EPI data acquisition at 100 μm×100 μm×200 μm spatial resolution with a 2 s effective repetition time (TR). The EPI-based T2* images acquired from head-fixed awake mice show little air-tissue interface-induced image distortion with the same spatial resolution as anatomical images (*Figure 2C and D*). Motion artifacts were detected at some time points of the fMRI time course, presenting large EPI image distortions (*Figure 3E*, *Video 2*), but can be removed using a censoring function during data analysis (*Figure 3—figure supplement 2*). These results have shown that the multi-slice 2D EPI enables brain-wide functional mapping of awake mice.

To map the brain function of awake mice with this high-resolution fMRI method, we first introduced a visual stimulation paradigm. Based on a block design regression analysis, we detected robust BOLD responses. Brain-wide functional maps using the visual stimulation paradigm were seen with activated areas highlighted along the visual pathways (*Figure 3*). These areas included the visual cortex (VC), superior colliculus (SC), lateral geniculate nucleus (LGN), and association cortex in the ACA. The ROI-specific localization was well characterized by overlapping the brain atlas and functional maps (*Figure 3C*). The ROI-based time courses demonstrated robust BOLD responses detected in awake mice (*Figure 3B*).

### Awake mouse fMRI with vibrissa stimulation

In contrast to a visual sensation, awake mice may flinch due to the sudden physical vibrissa stimulation causing severe motion artifacts during scanning. Prolonged training was needed to reduce motion artifacts during air-puff stimulation as shown in *Video 2* allowing for high-resolution fMRI of awake mice. The activated barrel cortex (BC) and ventroposterior medial nucleus (VPM) were seen related to stimulation of the contralateral whisker pad (time courses in *Figure 4B*, *Figure 3—figure supplement 3*). Brain-wide functional maps also showed activation in the motor cortex and a small portion

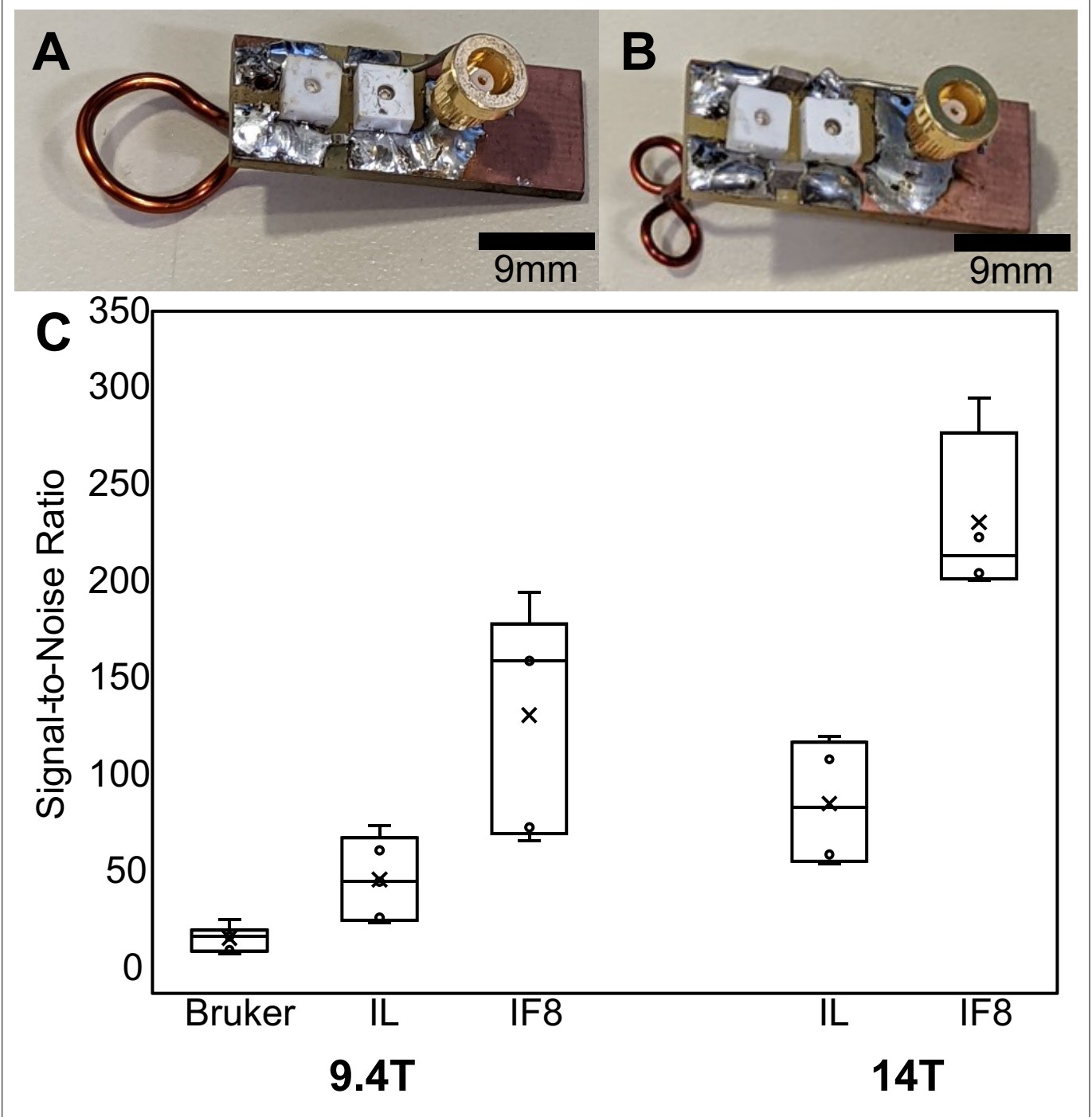

**Figure 1.** Comparison between implanted and commercial coils. (**A and B**) show representative (unattached) prototype coils in the single loop and 'figure 8' styles, respectively. (**C**) Box-and-whisker plot presents the cortical-specific signal-to-noise ratio (SNR) values calculated by dividing the mean signal of the upper cortex by the standard deviation of the noise to compare between commercial Bruker phased array surface coil, single loop implant, and 'figure 8' style implants. Bruker → commercial phased array coil, IL → implanted single loop coil, IF8 → implanted 'figure 8' coil. The bar graph shows the SNR of anatomical images acquired with different radio frequency (RF) coils using the 9.4 T scanner ($\overline{SNR}_{Bruker}$ = 27.2, N=6, $\overline{SNR}_{IL}$ = 57.5, N=5, $\overline{SNR}_{IF8}$ = 142.5, N=5) and the 14 T scanner ($\overline{SNR}_{IL}$ = 96.8, N=4, $\overline{SNR}_{IF8}$ = 209.2, N=5).

The online version of this article includes the following figure supplement(s) for figure 1:

**Figure supplement 1.** Schematic of circuit diagram of coil designed for [1]H imaging at 600 MHz.

**Figure supplement 2.** Comparison between implanted and commercial coils.

**Table 2.** Parts list for construction of 200 ¹H coils configured for 600 MHz (14 T).

| Part description | Cost |
| --- | --- |
| FR-4 PCB Chip, 0.4×0.8 in, 0.062 in thick | $214.00/200 |
| Capacitor 2.2 pF 2KV | $309.80/400 |
| Trimmer 0.6–2.5 pF | $1649.28/200 |
| Trimmer 5–18 pF | $1640.56/200 |
| Magnet wire 20AWG | $118.28/spool/710.8' |
| MCX socket | $1027.14/200 |
| Entire coil built by MRIBOT | $300/each |

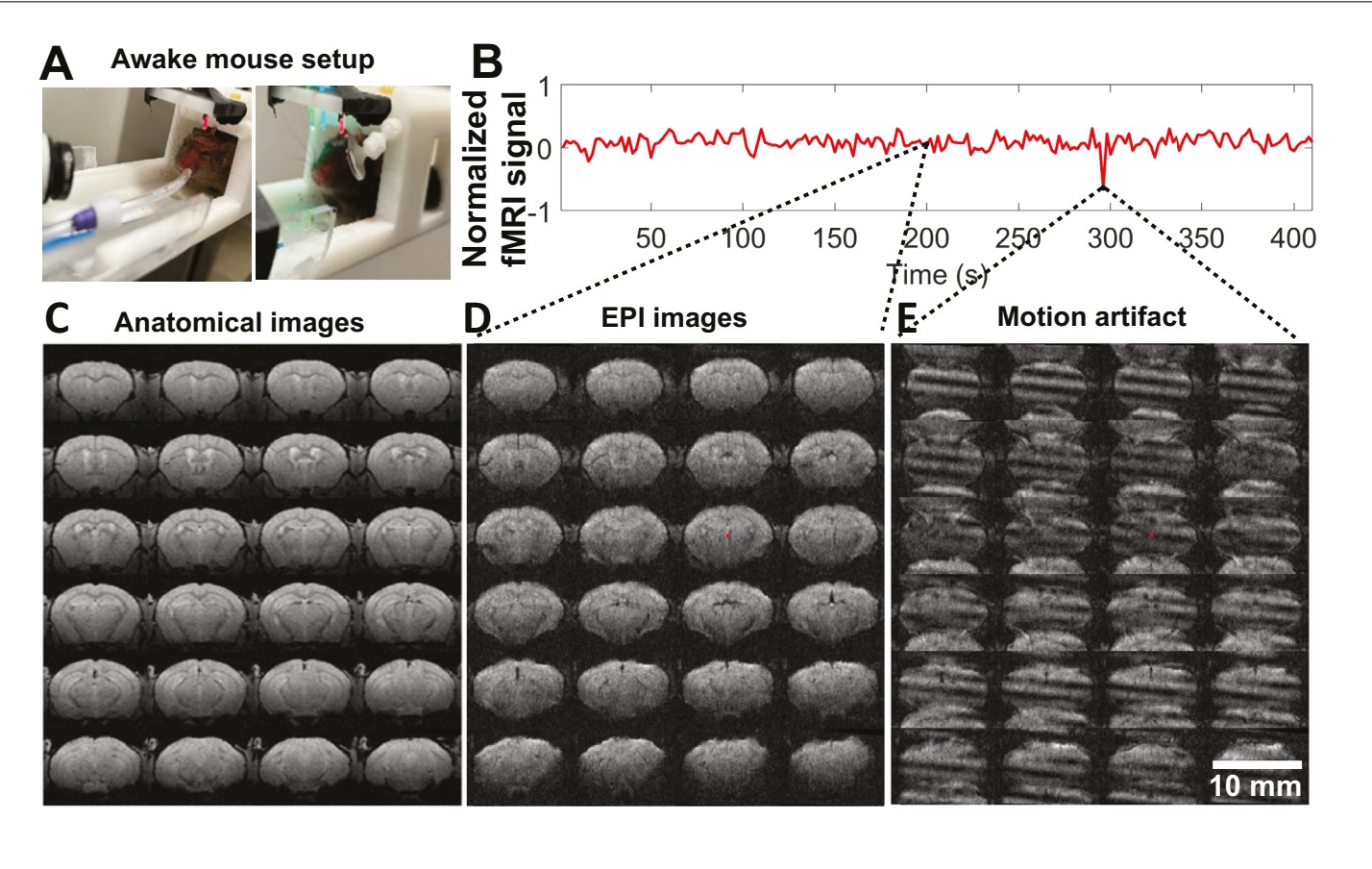

**Figure 2.** High-resolution awake mouse functional magnetic resonance imaging (fMRI) at 14 T. (**A**). The awake mouse setup with head-fixed position in a custom-built cradle for visual and vibrissa stimulation. (**B**) The representative fMRI time course of an awake mouse based on raw image data acquired from high-resolution echo planar imaging (EPI), enabling the trace of motion-induced artifacts. (**C**) The anatomical MRI images (fast low angle shot [FLASH]) acquired from one representative awake mouse, showing minimal susceptibility and whole brain coverage from the implanted surface coil. (**D**) The raw EPI fMRI image with same spatial resolution as the anatomical FLASH image. (**E**) The snapshot of the distorted images due to motion of the awake mouse during scanning. *Video 2* shows the video of motion-induced artifacts throughout the fMRI trial.

The online version of this article includes the following figure supplement(s) for figure 2:

**Figure supplement 1.** Results of acclimation training on animals.

**Figure supplement 2.** Figure showing motion data (.MOT file) of a representative mouse immediately following training (**A**) and after 1 month of scanning (**B**) indicating the amount of struggling the animal is doing during scanning.

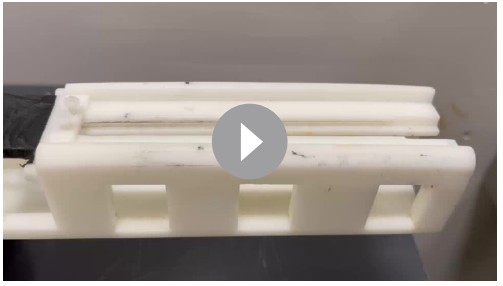

**Video 1.** Video illustrating how animals are set up through the cradle using the implanted radio frequency (RF) coil for awake mouse imaging.
https://elifesciences.org/articles/95528/figures#video1

of the ipsilateral BC. These results demonstrated the importance of distinguishing BOLD activation between external stimulation and voluntary movements while also confirming the feasibility to map brain-wide brain activations in awake behaving mice with 14 T fMRI.

## Prediction-related barrel cortical activity to patterned air-puff in awake mice

An interesting observation from the vibrissa stimulation was the activated VRA. The VRA only showed brief responses to air-puff in contrast to the typical duration of hemodynamic responses observed in the BC and VPM. This presents a good

**Figure 3.** Visual stimulation-evoked high-resolution functional magnetic resonance imaging (fMRI) of awake mice. (**A**) The brain-wide functional maps of awake mice show strong positive blood oxygen level-dependent (BOLD) activation in the visual cortex (VC), lateral geniculate nucleus (LGN), superior colliculus (SC), and anterior cingulate area (ACA) based on the group analysis. Highlighted brain regions are statistically generated using a one-way T-test with p<0.0001 (**B**) The averaged time course of the ROIs derived from the Allen Brain Atlas, demonstrating an evoked positive BOLD signal changes upon the 8 s visual stimulation (5 Hz 530 nm and 5.1 Hz 490 nm 20 ms light pluses). Each graph displays the average of 162 sets of 3 stimulation epochs. Shaded regions represent standard error. Red lines represent the 8 s stimulation duration. (**C**) Functional maps overlain with the brain atlas to highlight the activated brains regions: VC, SC, LGN, and ACA (N=13 (6F/7M)).

The online version of this article includes the following figure supplement(s) for figure 3:

**Figure supplement 1.** Figure showing stimulation paradigm in blue.

**Figure supplement 2.** The processing pipeline of the awake mouse functional magnetic resonance imaging (fMRI) datasets.

**Figure supplement 3.** Time course from single experiment showing strong blood oxygen level-dependent (BOLD) activations during stimulation paradigm.

**Video 2.** Video showing the real-time EPI raw images from awake mice. The real-time tracer from the selected point demonstrates the time points with motion, as well as the motion-induced image distortion during awake mouse fMRI.

https://elifesciences.org/articles/95528/figures#video2

landmark for studying higher-level processing of vibrissa sensation. Voxel-wise cross-correlation analysis was performed based on the VRA-specific fMRI dynamic changes. At a zero time shift (map developed from peak BOLD response), the VRA is strongly correlated with the hippocampus, cingulate cortex, and central thalamic regions (*Figure 5A*). Nevertheless, at a –6 s time shift (i.e. 2 s before stimulation onset), stronger correlation was observed at the contralateral BC, indicating anticipation of the repetitive air-puff in the block design (*Figure 5A*). To validate that this early BC activation was caused by learned anticipation of the time-fixed repetitive air-puff stimulation, we also analyzed the VRA-specific cross-correlation in a control group using a randomized stimulation paradigm. Although VRA remained strongly coupled with the other association cortices and subcortical regions at the zero time shift, no correlation was observed from the contralateral BC at the –6 s time shift (*Figure 5B*). The fMRI time course analysis from the contralateral BC also showed increased BOLD responses before the air-puff in the block design group, but not the randomized control group (*Figure 5C*). Quantitative analysis showed a significantly higher BOLD signal 2 s before the air-puff stimulation in the standard block design group when compared with the randomized group (*Figure 5D*). It should be noted that VRA responses between the two groups were similar, further confirming the anticipation-related early BC activation to repetitive air-puff stimulation.

## Discussion

In this study, we designed and implemented implantable RF coils for awake mouse fMRI, which also served as a headposts during data acquisition. Our design, based on previously published cable-free (inductive) RF coils (*Chen et al., 2022*), offered an easier pre-scan setup by eliminating the need to localize and secure the pickup coil for inductive coupling optimization. And while this current design showed reduced freedom for animal movement, implanted coils offer more stable sample loading and reduce the $B_0$ offset when compared to the previous version. This was also true when comparing to conventional RF coils as the motion of the animal would alter the loading and cause $B_1$ field variability during fMRI scanning.

### Technical considerations with awake mouse fMRI at 14 T

A few important factors should be considered to improve data quality using the implanted RF coils described in the present study. The first is animal motion. As this design was used for awake and minimally restrained mice, the animals would eventually move to adjust themselves (scratching, grooming, teeth grinding, etc.) during the scan. This will affect $B_0$ homogeneity and can cause ghosting if severe enough. This can be minimized through acclimation training which will also reduce unwanted stress. Other studies have animal restraint mechanisms that seek to restrain the body of the animal (*Desai et al., 2011*; *Chen et al., 2020*; *Madularu et al., 2017b*; *Harris et al., 2015*) but have the potential to cause unwanted stress which can affect the desired fMRI signals. Furthermore, $B_1$ variability was present through motion as well due to the current design of the coil. As the RF circuit chip sits above the animal's neck, body movement could alter the loading of the circuit, inducing $B_1$ artifacts through lifting or dropping the body toward or away from the circuit chip. Again, these artifacts can be minimized through proper training and stress reduction which was well accomplished for our study through the design and training method (*Xu et al., 2022*). Here, we see that while animal motion still exists (*Figure 2—figure supplement 2*) large struggling movements are minimized after the training method. Still, mice have a thin skull, which leads to the air-tissue interface being a non-negligible factor at ultra-high fields (e.g. 14 T). Therefore, the coil implantation shown here has reduced this source of inhomogeneity (*Figure 2—figure supplements 1 and 2*) and allowed for a consistent and

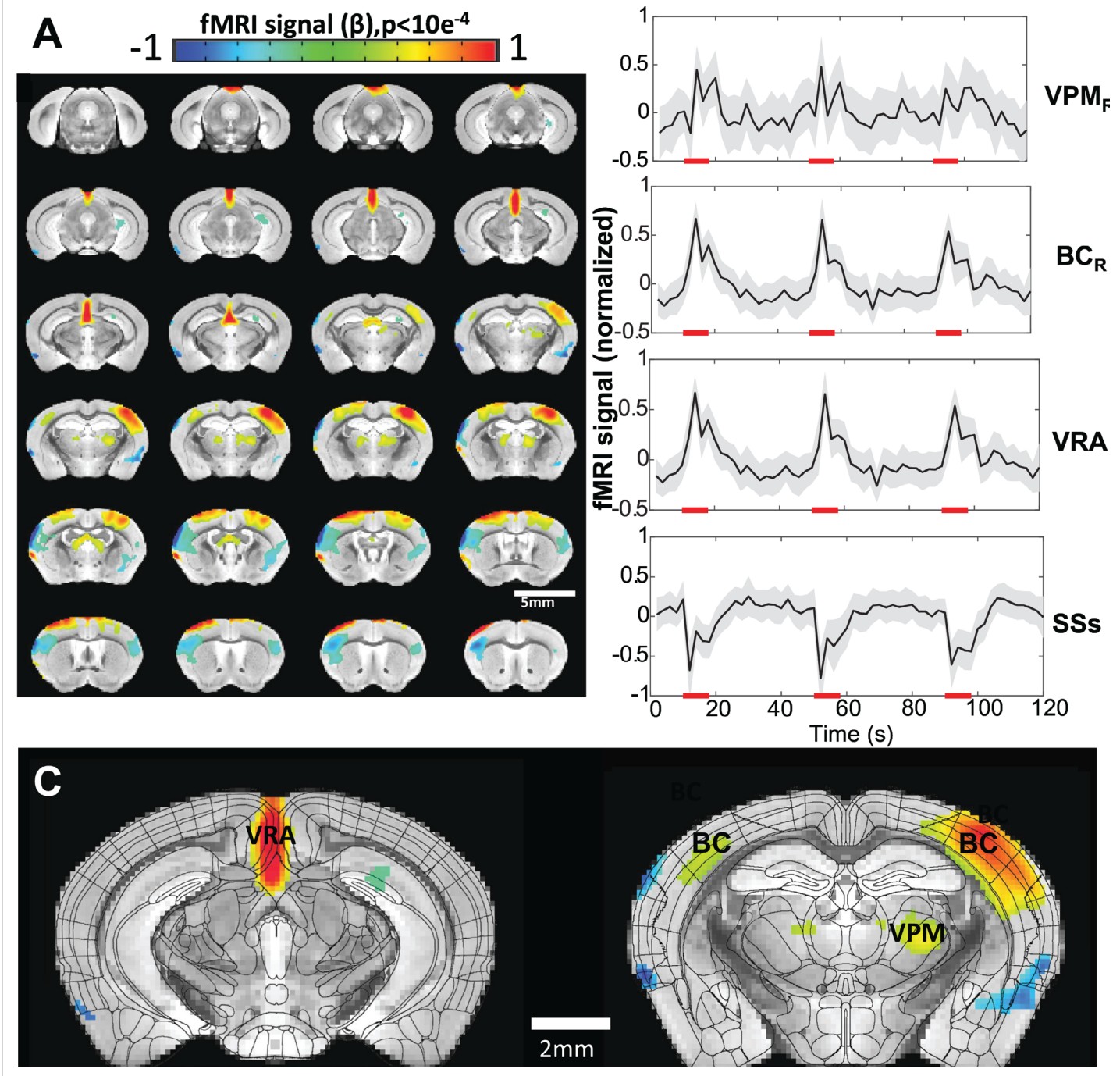

**Figure 4.** Vibrissa stimulation-evoked high-resolution functional magnetic resonance imaging (fMRI) of awake mice. (**A**) The brain-wide functional maps of awake mice show the strong positive blood oxygen level-dependent (BOLD) activation in the contralateral barrel cortex (BC) and ventral posteromedial nucleus (VPM) and posterior thalamic nucleus (PO). Positive BOLD signals are also detected at the motor cortex (MC) and the ventral retrosplenial area (VRA), as well as at the ipsilateral BC and thalamic nuclei. Negative BOLD signals are detected in supplementary somatosensory areas (SSs) (including nose and mouth) as well as part of the caudoputamen. Highlighted brain regions are statistically generated using a one-way T-test with p<0.0001. (**B**) The averaged time course based on the brain atlas ROIs for VMP, BC, and VRA, demonstrating positive BOLD signal changes upon the 8 s air-puff vibrissa stimulation (8 Hz, 10 ms). Averaged time course of the SSs ROI shows negative BOLD signal changes. Each graph displays the average of 279 sets of 3 stimulation epochs. Shaded regions represent standard error. Red lines represent the 8 s stimulation duration. (**C**) The functional maps are overlain with the brain atlas to highlight the activated vibrissa thalamocortical pathway (VPM→BC) and the VRA in awake mice (N=13 (6F/7M)).

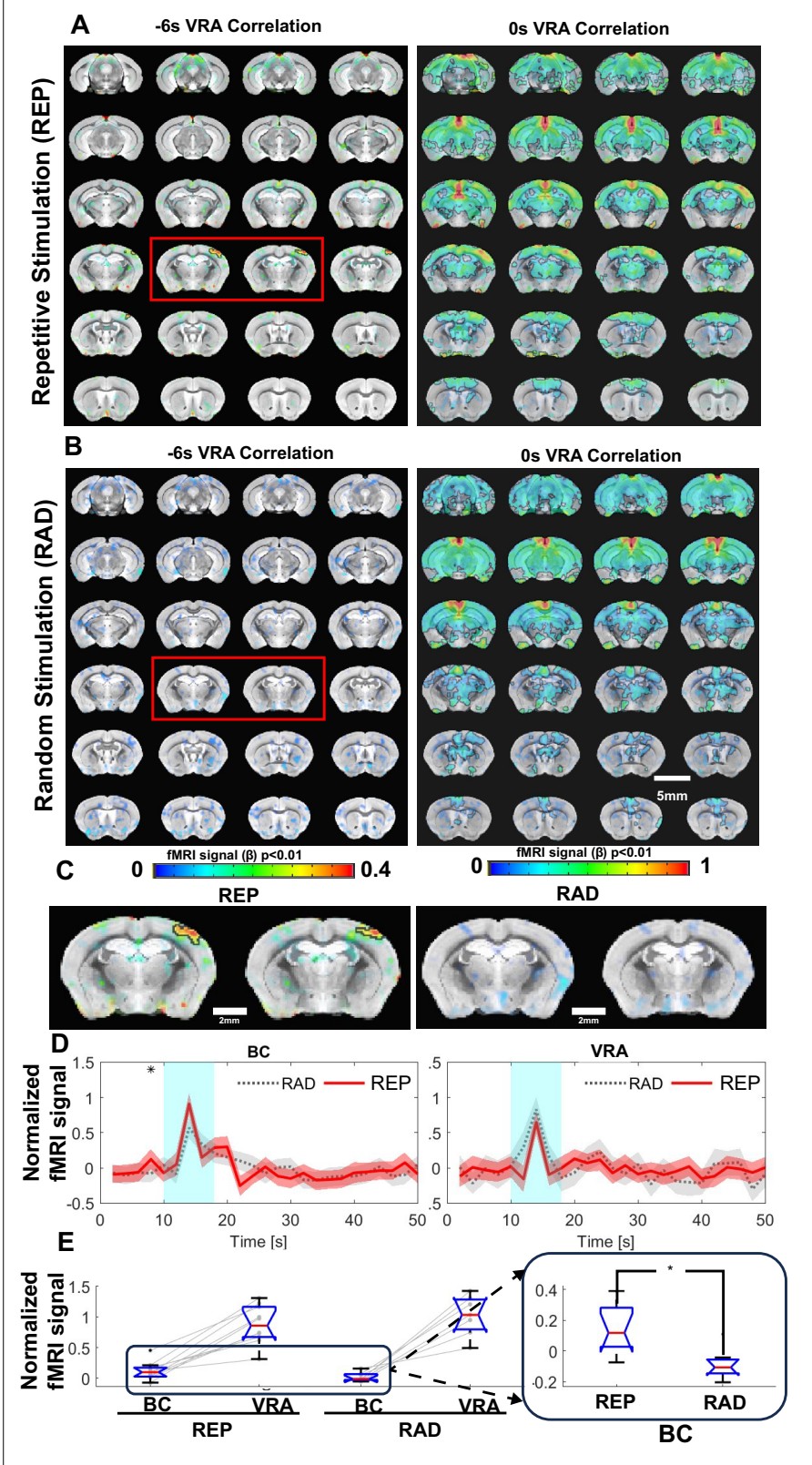

**Figure 5.** Ventral retrosplenial area (VRA)-based brain-wide correlation maps at different time shifts. (**A**) The VRA-based correlation maps at –6 s and 0 s time shifts of awake mice with repetitive stimulation (REP). The strong correlation in the contralateral barrel cortex (BC) is shown in the correlation map at the –6 s time shift (red box). Highlighted brain regions are statistically generated using a one-way T-test with p<0.01. (**B**) The VRA-based

*Figure 5 continued on next page*

*Figure 5 continued*

correlation maps at –6 s and 0 s time shifts of awake mice with randomized stimulation (RAD). No correlation is detected in the contralateral BC at the –6 s time shift (red box). Highlighted brain regions are statistically generated using a one-way T-test with p<0.01 (**C**) The enlarged images from the –6 s time shift correlation maps of REP and RAD groups, demonstrating the strong correlation patterns located at the contralateral BC only in the REP group. (**D**) The averaged time course from both contralateral BC and VRA of REP and RAD groups, showing that early positive blood oxygen level-dependent (BOLD) signals detected at 2 s prior to the stimulation in contralateral BC of the REP group and no significant difference detected in VRA. Shaded regions represent standard error. ✳ shows significance from two-tail two-way T-test (p<0.05). (**E**) The bar graph presents the mean BOLD signals of contralateral BC at 2 s prior to stimulation time point and peak signals of VRA in REP and RAD groups. The inset is the expanded bar graph to show the significantly higher BOLD signals detected in the contralateral BC at 2 s prior to stimulation in REP group using a two-tail two-way T-test (p<0.015, REP graph displays the average of 930 stimulation epochs, RAD graph displays the average of 240 stimulation epochs). (N=9 (4F/5M)).

stable shim. By implanting the coil on the surface of the skull, we can achieve a significantly higher SNR which can be comparable to cryoprobe designs at close distances. Moreso, this is the case with the 'figure 8' coil shown in this study which gives a five times increase in SNR over the standard commercially available four array mouse head coil for 9.4 T (both operating at room temperature) (*Figure 1*). This improvement allows much higher spatial resolution in awake mouse fMRI, at a two nanoliter voxel volume, compared to contemporary efforts in human brain mapping at sub-millimeter resolution (0.5–0.8 mm isotropic), a difference of two orders of magnitude (*Haenelt et al., 2023*; *Heidemann et al., 2012*; *Margalit et al., 2020*; *Feinberg et al., 2018*).

## The consideration of stress issues of awake mouse fMRI

Despite the extensive training procedure of the present work in comparison to the existing awake mouse fMRI studies (training strategies for awake mice fMRI have been reviewed by *Mandino et al., 2024* to show the overall training duration of existing studies), stress remains a confounding factor for the brain functional mapping in head-fixed mice. During animal training, we have measured both pupil dynamic and eye motion features from training sessions, both of which could be considered as potential surrogate of the stress levels of animals. It should be noted that stress may be related to increased frequency of eye blinking or twitching movements in human subjects (*Marcos-Ramiro et al., 2014*; *Haak et al., 2009*; *Del Carretto and Sessam, 2023*). However, the eyeblink of head-fixed mice has been used for behavioral conditioning to investigate motor learning in normal behaving mice (*Heiney et al., 2014*; *Chettih et al., 2011*; *Siegel et al., 2015*). Importantly, head-fixed mouse studies have shown that eye movements are significantly reduced compared to the free-moving mice (*Meyer et al., 2020*). The increased eye movement during the acclimation process would indicate an alleviated stress level of the head-fixed mice in our cases. Meanwhile, stress-related pupillary dilation could dominate the pupil dynamics at the early phase of training (*Zeng et al., 2022*). We have observed a gradually increased pupil dynamic power spectrum at the ultra-slow frequency during Phase 3, presenting the alleviated stress-related pupil dilation but recovered pupil dynamics to other factors, including arousal, locomotion, startles, etc. in normal behaving mice. Nevertheless, a recent study (*Juczewski et al., 2020*) shows that the corticosterone concentration in the blood samples of head-fixed mice is significantly reduced on day 25 following the training but remains higher than in the control mice. Also, the time-dependent changes of stress level during scanning could further confound the functional mapping results if longer than 1 hr. Thus, the impact of stress on brain functional mapping with awake mouse fMRI would need further investigation, of which the stress-related functional changes should not be neglected from the existing studies.

## Brain-wide functional mapping with visual and vibrissa stimulation

There are fMRI studies investigating the visual system in both anesthetized and awake mice (*Dinh et al., 2021*; *Zeng et al., 2022*; *Huang et al., 1996*; *Lee et al., 2019*; *Niranjan et al., 2016*; *Lungu et al., 2022*). In contrast to brain activation patterns at the VC, SC, and LGN (*Dinh et al., 2021*), robust ACA activation was also detected for awake mouse fMRI in this study (*Figure 3*). Since ACA has been closely involved in pupil dynamics, as well as arousal state regulation (*Ebitz and Platt, 2015*; *Joshi et al., 2016*; *Pfeffer et al., 2022*), the mapping of the ACA in awake mice during visual stimulation provides a meaningful way to validate the conscious state of mice during scanning. Similarly, there

are extensive rodent fMRI studies of vibrissa stimulation (*You et al., 2021*; *Van der Knaap et al., 2021*; *Lu et al., 2004*; *Balasco et al., 2022*; *Ferrier et al., 2020*; *Choi et al., 2023*). In contrast to the anesthetized state, this awake mouse fMRI detected not only activated contralateral BC and VPM, but also spread activation in the motor cortex, and small portion of the ipsilateral BC with positive BOLD signals. Although the air-puff stimulation was set and verified to deflect the whiskers of chosen side, videos of the mouse during scanning show that active bilateral whisking could be initiated upon air-puff. This could lead to bilateral activation of the motor cortex and the ipsilateral BC. Furthermore, studies have been performed to understand the transcallosal activity-mediated excitatory/inhibitory circuits by both fMRI and optical imaging (*Chen et al., 2022*; *Shim et al., 2020*; *Fujita et al., 2012*; *Grefkes et al., 2008*; *Lenzi et al., 2007*; *Reddy et al., 2000*). The potential transcallosal mediation of the negative BOLD signal detected in the superficial cortical area near BC will need to be further investigated. Also, these negative BOLD signals were detected across a large brain area, which is consistent with astrocyte-mediated negative BOLD during brain state changes reported in anesthetized rats (*Wang et al., 2018b*) and eye open/close-coupled arousal changes in unanesthetized monkeys (*Chang et al., 2016*). Although astrocytic $Ca^{2+}$ transients coincide with positive BOLD responses in the activated cortical areas, which align with the neurovascular coupling mechanism (*Takata et al., 2018*), there is emerging evidence to show that astrocytic $Ca^{2+}$ transients are coupled with both positive and negative BOLD responses in anesthetized rats (*Wang et al., 2018b*) and awake mice (*Tong et al., 2024*). An intriguing observation is that cortex-wide negative BOLD signals coupled with the spontaneous astrocytic $Ca^{2+}$ transients could co-exist with the positive BOLD signal detected at the activated cortex. Studies have shown that astrocytes are involved in regulating brain state changes (*Poskanzer and Yuste, 2016*), in particular, during locomotion (*Paukert et al., 2014*), and startle responses (*Srinivasan et al., 2015*). These brain state-dependent global negative BOLD responses are also related to the arousal changes of both non-human primates (*Chang et al., 2016*) and human subjects (*Setzer et al., 2022*). The established awake mouse fMRI platform with ultra-high spatial resolution will enable the brain-wide activity mapping of the functional nuclei contributing to the brain state changes of head-fixed awake mice in future studies.

Interestingly, vibrissa stimulation also led to robust VRA activation in awake mice (*Radwanska et al., 2010*). VRA serves as one of the major nodes of the default mode network across different species (*Lu et al., 2012*; *Andrews-Hanna et al., 2010*; *Vincent et al., 2007*; *Rilling et al., 2007*; *Raichle et al., 2001*). The vibrissa stimulation-evoked VRA activation suggests the higher-level cortical function contribute to vibrissa sensory processing in awake mice.

## VRA-coupled pre-stimulus BC activation in awake mice as a sign of anticipation

There are extensive studies investigating brain activation responsible for anticipation with fMRI and electrophysiological recordings (*Zhao et al., 2019*; *Martin et al., 2009*; *Zhao et al., 2022*; *Ploghaus et al., 2003*; *Sirotin and Das, 2009*). In contrast to the reward anticipation or audiovisual anticipation of naturalistic music and movie clips that demand more complex cognitive processing (*Pezzulo et al., 2007*; *McRobert et al., 2011*; *Burton et al., 2009*; *Ciesielski et al., 2012*), the repetitive air-puff stimulation delivered during head-fixed training for fMRI studies could serve as a simple paradigm to process the anticipatory responses in awake mice. Based on cross-correlation analysis with evoked VRA BOLD responses, the strongest correlation with the BC was detected from 6 s lag-time-based correlation maps, showing a positive BOLD signal at a time point 2 s prior to stimulus onset (*Figure 5*). This anticipatory BC response was not detected when the air-puff stimulation paradigm was randomized in another group of mice. VRA is known to be involved in prediction (*Smith et al., 2018*; *Miller et al., 2019*; *Auger and Maguire, 2013*) and has been coupled with temporal prediction in rodents (*Miller et al., 2019*; *Wyass and Van Groen, 1992*), as well as navigation efficiency involving spatial reference cues (*Miller et al., 2019*; *Auger and Maguire, 2013*; *Wyass and Van Groen, 1992*). Additionally, external somatosensory cues (e.g. the air-puff or brushing of whiskers) are an important factor when investigating prediction processing (*Taube, 2007*; *Valerio and Taube, 2012*; *Todd et al., 2019*; *Cooper et al., 2001*; *Keene and Bucci, 2021*). Previous work has shown that prediction of external stimulation will cause a hemodynamic response even in the absence of a stimulus (*Sirotin and Das, 2009*; *Yu et al., 2019*). In our study, we show that after continued regularly spaced stimulation, early somatosensory hemodynamic responses begin to have a significant impact seen in the averaged

BOLD response time course. These anticipatory hemodynamic responses are a result of the continuous training for mice experiencing months of repetitive stimulation. The increased BOLD signal in the BC before the stimulus onset shows strong cross-correlation to the VRA activation, but VRA activation is not dependent on the pre-stimulus activation in the BC. This can be seen through the comparable VRA BOLD responses between repetitive and randomized air-puff stimulation paradigms (*Figure 5C*). This result indicates that VRA response mediates external sensory perception and may serve as a key association cortical area for the processing of the anticipated vibrissa signals but is not solely dependent on the prediction of incoming stimulus.

## Acknowledgements

The research benefited from funding from the NIH Brain Initiative grants (RF1NS113278, RF1NS124778, R01NS122904, R01NS120594, and R21NS121642), U19 Cooperative Agreement Grant (U19NS123717), S10 instrument grants (S10OD028616 and S10RR025563) to the MGH/Harvard-MIT Program in Health Sciences and Technology Martinos Center, and NSF CBET grant (2123970).

## Additional information

### Competing interests

Xin Yu: Cofounder of MRIBOT LLC. The other authors declare that no competing interests exist.

### Funding

| Funder | Grant reference number | Author |
| --- | --- | --- |
| National Institute of Neurological Disorders and Stroke | RF1NS113278 | Xin Yu |
| National Institute of Neurological Disorders and Stroke | RF1NS124778 | Xin Yu |
| National Institute of Neurological Disorders and Stroke | R01NS122904 | Xin Yu |
| National Institute of Neurological Disorders and Stroke | R01NS120594 | Xin Yu |
| National Institute of Neurological Disorders and Stroke | R21NS121642 | Xin Yu |
| National Institute of Neurological Disorders and Stroke | U19NS123717 | Anna Devor |
| NIH Office of the Director | S10OD028616 | Xin Yu |
| National Center for Research Resources | S10RR025563 | Xin Yu |
| National Science Foundation | 2123970 | Xin Yu |

The funders had no role in study design, data collection and interpretation, or the decision to submit the work for publication.

### Author contributions

David Hike, Xiaochen Liu, Conceptualization, Resources, Data curation, Formal analysis, Validation, Investigation, Methodology, Writing – original draft, Project administration, Writing – review and editing; Zeping Xie, Data curation, Investigation, Methodology; Bei Zhang, Resources, Validation, Methodology; Sangcheon Choi, Software, Formal analysis; Xiaoqing Alice Zhou, Formal analysis,

Methodology; Andy Liu, Resources, Data curation; Alyssa Murstein, Resources, Formal analysis; Yuanyuan Jiang, Formal analysis; Anna Devor, Resources, Funding acquisition, Writing – review and editing; Xin Yu, Conceptualization, Data curation, Supervision, Funding acquisition, Methodology, Writing – review and editing

## Author ORCIDs
David Hike ⓘ http://orcid.org/0000-0001-5294-1767
Xiaochen Liu ⓘ http://orcid.org/0000-0001-6342-7704
Sangcheon Choi ⓘ http://orcid.org/0000-0001-7327-1344
Andy Liu ⓘ http://orcid.org/0009-0004-0080-8429
Yuanyuan Jiang ⓘ http://orcid.org/0000-0001-7758-7450
Anna Devor ⓘ https://orcid.org/0000-0002-5143-3960
Xin Yu ⓘ https://orcid.org/0000-0001-9890-5489

## Ethics
All animal procedures were conducted in accordance with protocols approved by the Massachusetts General Hospital (MGH) Institutional Animal Care and Use Committee (IACUC), and animals were cared for according to the requirements of the National Research Council's Guide for the Care and Use of Laboratory Animals.

Reviewer #1 (Public review): https://doi.org/10.7554/eLife.95528.3.sa1
Reviewer #2 (Public review): https://doi.org/10.7554/eLife.95528.3.sa2
Author response https://doi.org/10.7554/eLife.95528.3.sa3

# Additional files

## Supplementary files
MDAR checklist

## Data availability
Data is available for download from OpenNeuro: Whisker (https://doi.org/10.18112/openneuro.ds005496.v1.0.1), Visual (https://doi.org/10.18112/openneuro.ds005497.v1.0.0) and Zenodo:SNR Line Profile Data & Data Processing Scripts: (https://zenodo.org/doi/10.5281/zenodo.13821455).

The following datasets were generated:

| Author(s) | Year | Dataset title | Dataset URL | Database and Identifier |
|---|---|---|---|---|
| Hike D, Liu X, Xie Z, Zhang B, Choi S, Zhou XA, Liu A, Murstein A, Jiang Y, Devor A, Yu X | 2024 | Whisker Stim | https://openneuro.org/datasets/ds005496/versions/1.0.1 | OpenNeuro, ds005496 |
| Liu X, Hike D, Choi S, Man W, Ran C, Zhou XA, Jiang Y, Yu X | 2024 | Visual Stim | https://openneuro.org/datasets/ds005497/versions/1.0.0 | OpenNeuro, ds005497 |
| Hike D, Liu X, Yu X | 2024 | Awake Mouse | https://doi.org/10.5281/zenodo.13821456 | Zenodo, 10.5281/zenodo.13821456 |

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
