## [Editor Report · eLife Assessment]

This is a **valuable** study describing an implementation of awake mouse fMRI with implanted head coils at high fields. The evidence presented is **convincing**, combining technical advances with interesting neuroscience applications showing that mice anticipate stimuli given at regular (but at irregular) intervals.

---

## [Referee Report · Reviewer #1 (Public review)]

Summary:

The authors bring together implanted radiofrequency coils, high-field MRI imaging, awake animal imaging, and sensory stimulation methods in a technological demonstration. The results are very detailed descriptions of the sensory systems under investigation.

Strengths:

The maps are qualitatively excellent for rodent whole-brain imaging.

The design of the holder and the coil is pretty clever.

Weaknesses:

Some unexpected regions appear on the whole brain maps, and the discussion of these regions is succinct.

The authors do not make the work and effort to train the animals and average the data from several hundred trials apparent enough. This is important for any reader who would like to consider implementing this technology.

The data is not available. This does not let the readers make their own assessment of the results.

Comments on revisions:

All good, I can but only congratulate the authors on a study well done.

---

## [Referee Report · Reviewer #2 (Public review)]

This work explores the advancement of awake mouse BOLD-fMRI at 14 Tesla. The study introduces custom-implanted RF coils aimed at improving signal-to-noise ratio (SNR) and assesses their performance in detecting responses to stimuli in awake mice. The coils show significant SNR improvements and are a noteworthy innovation. Detailed descriptions of the coil design, including parts lists and diagrams, enhance the reproducibility of the methods. A thorough 5-week acclimation protocol was used to minimize stress and motion during imaging. Stress was primarily evaluated using eye tracking which, in an fMRI setting, is novel and could help move the field forward with further validation (within the context of fMRI experiments). Overall, the authors successfully demonstrate high-resolution awake mouse fMRI with enhanced SNR; thus achieving their primary aim.

This work is likely to significantly impact the field by demonstrating the feasibility of high-quality awake mouse fMRI, potentially leading to more accurate and artifact-free studies of brain function. The detailed methods shared will facilitate replication and adoption by other researchers, promoting standardized practices. The methods and data provided serve as valuable resources for the neuroscience community.

---

## [Author Response]

The following is the authors’ response to the original reviews.

**Public Reviews:**

**Reviewer #1 (Public Review):**
Summary:The authors bring together implanted radiofrequency coils, high-field MRI imaging, awake animal imaging, and sensory stimulation methods in a technological demonstration. The results are very detailed descriptions of the sensory systems under investigation.Strengths:- The maps are qualitatively excellent for rodent whole-brain imaging. - The design of the holder and the coil is pretty clever.Weaknesses:- Some unexpected regions appear on the whole brain maps, and the discussion of these regions is succinct.- The authors do not make the work and effort to train the animals and average the data from several hundred trials apparent enough. This is important for any reader who would like to consider implementing this technology.- The data is not available. This does not let the readers make their own assessment of the results.

Thank you for the comments on this manuscript. We have provided more detailed discussion of the unexpected regions(page 18 – line 491-494) and training procedures(page7-9 – line 172-236). We also uploaded the datasets to OpenNeuro

Whisker (https://doi.org/10.18112/openneuro.ds005496.v1.0.1), Visual (https://doi.org/10.18112/openneuro.ds005497.v1.0.0) and Zenodo:

SNR Line Profile Data & Data Processing Scripts: (https://zenodo.org/doi/10.5281/zenodo.13821455).

**Reviewer #2 (Public Review):**
Summary:The manuscript by Hike et al. entitled 'High-resolution awake mouse fMRI at 14 Tesla' describes the implementation of awake mouse BOLD-fMRI at high field. This work is timely as the field of mouse fMRI is working toward collecting high-quality data from awake animals. Imaging awake subjects offers opportunities to study brain function that are otherwise not possible under the more common anesthetized conditions. Not to mention the confounding effects that anesthesia has on neurovascular coupling. What has made progress in this area slow (relative to other imaging approaches like optical imaging) is the environment within the MRI scanner (high acoustic noise) - as well as the intolerance of head and body motion. This work adds to a relatively small, but quickly growing literature on awake mouse fMRI. The findings in the study include testing of an implanted head-coil (for MRI data reception). Two designs are described and the SNR of these units at 9.4T and 14T are reported. Further, responses to visual as well as whisker stimulation recorded in acclimated awake mice are shown. The most interesting finding, and most novel, is the observation that mice seem to learn to anticipate the presentation of the stimulus - as demonstrated by activations evident ~6 seconds prior to the presentation of the stimulus when stimuli are delivered at regular intervals (but not when stimuli are presented at random intervals). These kinds of studies are very challenging to do. The surgical preparation and length of time invested into training animals are grueling. I also see this work as a step in the right direction and evidence of the foundations for lots of interesting future work. However, I also found a few shortcomings listed below.Weaknesses:(1) The surface coil, although offering a great SNR boost at the surface, ultimately comes at a cost of lower SNR in deeper more removed brain regions in comparison to commercially available Bruker coils (at room temperature). This should be quantified. A rough comparison in SNR is drawn between the implanted coils and the Bruker Cryoprobe - this should be a quantitative comparison (if possible) - including any differences in SNR in deeper brain structures. There are drawbacks to the Cryoprobe, which can be discussed, but a more thorough comparison between the implanted coils, and other existing options should be provided the Cryoprobe has been used previously in awake mouse experiments (Sensory evoked fMRI paradigms in awake mice - Chen, Physiological effects of a habituation procedure for functional MRI in awake mice using a cryogenic radiofrequency probe – Yoshida, PREVIOUS REFERENCE). Further, the details of how to build the implanted coils should be provided (shared) - this should include a parts list as well as detailed instructions on how to build the units. Also, how expensive are they? And can they be reused?

Thank you for the comment. We did not use a Bruker Cryoprobe for this work but rather a Bruker 4array surface coil. We are unable to compare to a cryoprobe since we do not have access to one for our system. A comparison to previously published data using different scanners could be possible but would require the sequence contain identical parameters to avoid introducing an uncontrollable variable, we are planning to recruit different laboratories to test the implanted RF coils with their existing cryoprobes in the future study.

We have included an updated figure comparing SNR at different depths across the Bruker 4-array coil and the implanted RF coils. As shown in Figure 1—figure supplement 2, there is significant SNR enhancement up to 4 mm cortical depth for both single loop and ‘figure 8’ implanted RF coils in comparison to the Bruker 4-array coil.

**Author response image 1. sa3fig1:** Comparison between implanted and commercial coils. (**A**) shows representative coils in the single loop (left) and ‘figure 8’ styles (right). Supplementary Table 1 provides a parts list and cost for making these coils and Supplementary Figure 1 provides a circuit diagram to assemble. (**B**) presents the SNR line profile values as a function of distance from Pia Matter for each coil tested at 9.4T: commercial phased array surface coil (4 Array), implanted single loop coil, and implanted ‘figure 8’ coil. SNR values were calculated by dividing the signal by the standard deviation of the noise. (**C-E**) show a representative FLASH image with line profile of SNR measurements from each of the coils used to create the graph seen in **B**. Clear visual improvement in SNR can be seen in figures **C-E**. C – Commercial phased array. D – Single loop at 9.4T. E – Figure 8 at 9.4T. (N4 array = 6, Nsingle loop = 5, Nfigure 8 = 5)

Additionally, we have added a supplementary figure (Figure 1—figure supplement 1) of a circuit diagram, in an effort to disseminate the prototype design of the coils to other laboratories. We have included a detailed parts list with the cost for construction of the coils configured for our scanner(supp table 1). These specifics though would need to be adjusted to the precise field strength/bore size/animal the coil was being built for. As for reusability, the copper wire is cemented to the animal skull and this implantable coil should be considered as consumables for the awake mouse experiments, though the PCB parts can be retrieved.

(2) In the introduction, the authors state that "Awake mouse fMRI has been well investigated". I disagree with this statement and others in the manuscript that gives the reader the impression that awake experiments are not a challenging and unresolved approach to fMRI experiments in mice (or rodents). Although there are multiple labs (maybe 15 worldwide) that have conducted awake mouse experiments (with varying degrees of success/thoroughness), we are far from a standardized approach. This is a strength of the current work and should be highlighted as such. I encourage the authors to read the recent systematic review that was published on this topic in Cerebral Cortex by Mandino et al. There are several elements in there that should influence the tone of this piece including awake mouse implementations with the Bruker Cryoprobe, prevalence of surgical preparations, and evaluations of stress.

Thank you for the comment. We agree with the reviewer that the current stage of awake mouse fMRI studies remains to be improved. And, we have revised the Introduction to highlight the state-of-theart of awake mouse fMRI (Page 4 – line 81-88).

(3) The authors also comment on implanted coils reducing animal stress - I don't know where this comment is coming from, as this has not been reported in the literature (to my knowledge) and the authors don't appear to have evaluated stress in their mice.

Since question 3 and 4 are highly related to the acclimation procedures, we will answer the two questions together.

(4) Following on the above point, measures of motion, stress, and more details on the acclimation procedure that was implemented in this study should be included.

We thank the reviewer to raise the animal training issues.

During the animal training, we have measured both pupil dynamic and eye motion features from training sessions, of which the detailed procedure is described in Methods (page 7-9 – line 172-236).

The training procedure is carried out over a total of 5 weeks with four phases of training: i. Holding animal in hands, ii. Head-fixation and pupillometry, iii. Head-fixation and pupillometry with mockMRI acoustic exposure, iv. Head-fixation and pupillometry with Echo-Planar-Imaging (EPI) in the MR scanner. Author response table 1

**Author response table 1. sa3table1:** 

Training Day of Each Phase	Phase 1 (hold in hand)	Phase 2 (holder +pupil)	Phase 3 (Mock-MRI+pupil)	Phase 4 (EPI +pupil)
1	5 mins	15 mins	30 mins	60 mins (rs)
2	10 mins	30 mins	30 mins	60 mins (rs)
3			30 mins	60 mins (stim)
4			30 mins	60 mins (stim)
5			60 mins	
6			60 mins	
7			60 mins	
8			60 mins	

As shown in Author response image 2, the spectral power of pupil dynamics (<0.02Hz) and eye movements gradually increased as a function of the training time for head-fixed mice exposed to the mock MRI acoustic environment during phase 3. In phase 4, when head-fixed mice were put into the scanner for the first time, both eye movements and pupil dynamics were initially reduced during scanning but recovered to an acclimated state on Day 2, similar to the level on Day 8 of phase 3. These behavioral outputs would provide an alternative way to monitor the stress levels of the mice.

**Author response image 2. sa3fig2:** The eye movements (**A**) and power spectra of pupil dynamics (<0.02Hz) (**B**) change during different training phases.

It should be noted that stress may be related to increased frequency of eye blinking or twitching movements in human subjects(1–3). Whereas, the eyeblink of head-fixed mice has been used for behavioral conditioning to investigate motor learning in normal behaving mice(4–6). Importantly, head-fixed mouse studies have shown that eye movements are significantly reduced compared to the free-moving mice(7). The increased eye movement during acclimation process would indicate an alleviated stress level of the head-fixed mice in our cases. Meanwhile, stress-related pupillary dilation could dominate the pupil dynamics at the early phase of training(8). We have observed a gradually increased pupil dynamic power spectrum at the ultra-slow frequency during phase 3, presenting the alleviated stress-related pupil dilation but recovered pupil dynamics to other factors, including arousal, locomotion, startles, etc. in normal behaving mice. Despite the extensive training procedure of the present work in comparison to the existing awake mouse fMRI studies (training strategies for awake mice fMRI have been reviewed by Mandino et al. to show the overall training duration of existing studies(9)), the stress remains a confounding factor for the brain functional mapping in head-fixed mice. In particular, a recent study(10) shows that the corticosterone concentration in the blood samples of head-fixed mice is significantly reduced on Day 25 following the training but remains higher than in the control mice. In the discussion section, we have discussed the potential issues of stress-related confounding factors for awake mouse fMRI studies (Page 16 – lines 436-458).

(1) A. Marcos-Ramiro, D. Pizarro-Perez, M. Marron-Romera, D. Gatica-Perez, Automatic blinking detection towards stress discovery. ICMI 2014 - Proceedings of the 2014 International Conference on Multimodal Interaction 307–310 (2014). https://doi.org/10.1145/2663204.2663239/SUPPL_FILE/ICMI1520.MP4.

(2) M. Haak, S. Bos, S. Panic, L. Rothkrantz, DETECTING STRESS USING EYE BLINKS AND BRAIN ACTIVITY FROM EEG SIGNALS. Lance 21, 76 (2009).

(3) E. Del Carretto Di Ponti E Sessam, Exploring the impact of Stress and Cognitive Workload on Eye Movements: A Preliminary Study. (2023).

(4) S. A. Heiney, M. P. Wohl, S. N. Chettih, L. I. Ru olo, J. F. Medina, Cerebellar-dependent expression of motor learning during eyeblink conditioning in head-fixed mice. J Neurosci 34, 14845–14853 (2014).

(5) S. N. Chettih, S. D. Mcdougle, L. I. Ruffolo, J. F. Medina, Adaptive timing of motor output in the mouse: The role of movement oscillations in eyelid conditioning. Front Integr Neurosci 5, 12996 (2011).

(6) J. J. Siegel, et al., Trace Eyeblink Conditioning in Mice Is Dependent upon the Dorsal Medial Prefrontal Cortex, Cerebellum, and Amygdala: Behavioral Characterization and Functional Circuitry. eNeuro 2, 51–65 (2015).

(7) A. F. Meyer, J. O’Keefe, J. Poort, Two Distinct Types of Eye-Head Coupling in Freely Moving Mice. Current Biology 30, 2116-2130.e6 (2020).

(8) H. Zeng, Y. Jiang, S. Beer-Hammer, X. Yu, Awake Mouse fMRI and Pupillary Recordings in the UltraHigh Magnetic Field. Front Neurosci 16, 886709 (2022).

(9) F. Mandino, S. Vujic, J. Grandjean, E. M. R. Lake, Where do we stand on fMRI in awake mice? Cereb Cortex 34 (2024).

(10) K. Juczewski, J. A. Koussa, A. J. Kesner, J. O. Lee, D. M. Lovinger, Stress and behavioral correlates in the head-fixed method: stress measurements, habituation dynamics, locomotion, and motor-skill learning in mice. Scientific Reports 2020 10:1 10, 1–19 (2020).

(5) It wasn't clear to me at what times the loop versus "Figure 8" coil was being used, nor how many mice (or how much data) were included in each experiment/plot. There is also no mention of biological sex.

Thank you for the comment. We have clarified sex and number. The ‘figure 8’ coil was only used as part of development to show the improvement of the coil design for cortical measurements. The detailed information is described in Method (Page 6 – line 127-129 & Page 10 – line 269-270). Additionally animal numbers have been included in the figure captions.

(6) Building on the points above, the manuscript overall lacks experimental detail (especially since the format has the results prior to the methods).

Thank you for the comment. We have modified the manuscript to increase the experimental detail and moved the methods section before the results.

(7) An observation is made in the manuscript that there is an appreciable amount of negative BOLD signal. The authors speculate that this may come from astrocyte-mediated BOLD during brain state changes (and cite anesthetized rat and non-human primate experiments). This is very strange to me. First, the negative BOLD signal is not plotted (please do this), further, there are studies in awake mice that measure astrocyte activation eliciting positive BOLD responses (see Takata et al. in Glia, 2017).

We thank the reviewer to raise the negative BOLD fMRI observation issue. We added a subplot of the negative BOLD signal changes in the revised Figure 4. This negative BOLD signals across cortical areas could be coupled with brain state changes upon air-puff-induced startle responses. Our future studies are focusing on elucidating the brain-wide activity changes of awake mice with fMRI. We also provide a detailed discussion of the potential mechanism underlying the negative BOLD fMRI signals. First, as reported in the paper (suggested by the reviewer), astrocytic Ca2+ transients coincide with positive BOLD responses in the activated cortical areas, which is aligning with the neurovascular coupling (NVC) mechanism. However, there is emerging evidence to show that astrocytic Ca2+ transients are coupled with both positive and negative BOLD responses in anesthetized rats(11) and awake mice(12). An intriguing observation is that cortex-wide negative BOLD signals coupled with the spontaneous astrocytic Ca2+ transients could co-exist with the positive BOLD signal detected at the activated cortex. Studies have shown that astrocytes are involved in regulating brain state changes(13), in particular, during locomotion(14) and startle responses(15). These brain state-dependent global negative BOLD responses are also related to the arousal changes of both non-human primates(16) and human subjects(17). The established awake mouse fMRI platform with ultra-high spatial resolution will enable the brain-wide activity mapping of the functional nuclei contributing to the brain state changes of head-fixed awake mice in future studies. (Page 17-18 – Line 478-490)

(11) M. Wang, Y. He, T. J. Sejnowski, X. Yu, Brain-state dependent astrocytic Ca2+ signals are coupled to both positive and negative BOLD-fMRI signals. Proc Natl Acad Sci U S A 115, E1647–E1656 (2018).

(12) C. Tong, Y. Zou, Y. Xia, W. Li, Z. Liang, Astrocytic calcium signal bidirectionally regulated BOLD-fMRI signals in awake mice in Proc. Intl. Soc. Mag. Reson. Med. 32, (2024).

(13) K. E. Poskanzer, R. Yuste, Astrocytes regulate cortical state switching in vivo. Proc Natl Acad Sci U S A 113, E2675–E2684 (2016).

(14) M. Paukert, et al., Norepinephrine controls astroglial responsiveness to local circuit activity. Neuron 82, 1263–1270 (2014).

(15) R. Srinivasan, et al., Ca2+ signaling in astrocytes from IP3R2−/− mice in brain slices and during startle responses in vivo. Nat Neurosci 18, 708 (2015).

(16) C. Chang, et al., Tracking brain arousal fluctuations with fMRI. Proc Natl Acad Sci U S A 113, 4518– 4523 (2016).

(17) B. Setzer, et al., A temporal sequence of thalamic activity unfolds at transitions in behavioral arousal state. Nat Commun 13 (2022).

**Recommendations for the authors:**

**Reviewer #1 (Recommendations For The Authors):**
I really enjoyed this work. The maps shown are among the best-quality maps out there. Here are suggestions to the authors.(1) Both the ACA and VRA are rather unexpected. The authors explain these briefly as being part of the associative cortical areas. Both the ACA and VRA are not canonical associative areas (or at least not to us). This warrants a stronger discussion.

To verify both ACA and VRA as associate areas, we provide the connectivity map projections from the Allen Brain Atlas (seen below). These projections are derived from a Cre-dependent AAV tracing of axonal projections. We have included an explanation of this in the introduction.

**Author response image 3. sa3fig3:** Representative images are shown indicating connections between the barrel cortex and retrosplenial area from an injection in the barrel cortex (Left panel) as well as the visual cortex and cingulate connection from an injection in the visual cortex (Right panel). Images are of connectivity map projections from the Allen Brain Atlas derived from a Cre-dependent AAV tracing of axonal projections (94)

(2) This is a lot of work. But looking at the figures, this is not obvious. We read in the caption that several hundred trials were used. It would be good to also specify how many mice. It would be clearer to represent this info in the figure as well to support the fact that this is not a trivial acquisition.

Thank the reviewer to raise the effort issue. We have edited the figure to include this information and included the numbers in the text as well

(3) The training protocol is seemingly extensive, but this is only visible by following another reference. Including a description in this work would help the reader make sense of the effort that went into this work.

We thank the reviewer to raise the training protocol issue. We have more thoroughly discussed the training method used for this study (page 7-9 – line 172-236)

(4) I really would love to see that dataset made freely available - this should be the norm.

The datasets have been uploaded to OpenNeuro

Whisker (https://doi.org/10.18112/openneuro.ds005496.v1.0.1), Visual (https://doi.org/10.18112/openneuro.ds005497.v1.0.0) and Zenodo:

SNR Line Profile Data & Data Processing Scripts:

(https://zenodo.org/doi/10.5281/zenodo.13821455).

(page 21 – line 573-579)

**Reviewer #2 (Recommendations For The Authors):**
(1) I'm a little confused about the stimulation paradigm and the effect of it causing an effective 2second TR (which is on the long side) - please elaborate (a figure might be helpful). The paradigm for visual stimulation also seems elaborate, can you please explain the logic and how it was developed?

Thank you for raising the detailed stimulation paradigm issues. The stimulation paradigm is independent and does not interfere with the setup of the effective 2-second TR. The 2-second TR is based on the usage of 2-segment EPI, each with a TR of 1-second. The application of 2-segment paradigm enables the echo spacing with 0.52 ms with effective image bandwidth with 3858Hz, assuring less image distortion. The stimulation paradigm was defined by an “8s on, 32s off” epoch such to elicit a strong BOLD response and could be used for any reasonable TR duration.

We have included a figure outlining the stimulation paradigm (Figure 3—figure supplement 1)

(2) I had difficulties viewing the movies (on my MAC).

Thank you for this note. We have re-upload the videos in .mov format